# Association between industry payments and prescriptions of long-acting insulin: An observational study with propensity score matching

**Kosuke Inoue**[1,2]*, **Yusuke Tsugawa**[3,4], **Carol M. Mangione**[3,4], **O. Kenrik Duru**[3]

**1** Department of Epidemiology, UCLA Fielding School of Public Health, Los Angeles, California, United States of America, **2** Department of Social Epidemiology, Graduate School of Medicine, Kyoto University, Kyoto, Japan, **3** Division of General Internal Medicine and Health Services Research, David Geffen School of Medicine at UCLA, Los Angeles, California, United States of America, **4** Department of Health Policy and Management, UCLA Fielding School of Public Health, Los Angeles, California, United States of America

* koinoue@ucla.edu

## Abstract

### Background

The rapidly increased spending on insulin is a major public health issue in the United States. Industry marketing might be one of the upstream determinants of physicians' prescription of long-acting insulin—the most commonly used and costly type of insulin, but the evidence is lacking. We therefore aimed to investigate the association between industry payments to physicians and subsequent prescriptions of long-acting insulin.

### Methods and findings

Using the databases of Open Payments and Medicare Part D, we examined the association between the receipt of industry payments for long-acting insulin in 2016 and (1) the number of claims; (2) the costs paid for all claims; and (3) the costs per claim of long-acting insulin in 2017. We also examined the association between the receipt of payments and the change in these outcomes from 2016 to 2017. We employed propensity score matching to adjust for the physician-level characteristics (sex, years in practice, specialty, and medical school attended). Among 145,587 eligible physicians treating Medicare beneficiaries, 51,851 physicians received industry payments for long-acting insulin worth $22.3 million. In the propensity score–matched analysis including 102,590 physicians, we found that physicians who received the payments prescribed a higher number of claims (adjusted difference, 57.8; 95% CI, 55.8 to 59.7), higher costs for total claims (adjusted difference, +$22,111; 95% CI, $21,387 to $22,836), and higher costs per claim (adjusted difference, +$71.1; 95% CI, $69.0 to $73.2) of long-acting insulin, compared with physicians who did not receive the payments. The association was also found for changes in these outcomes from 2016 to 2017. Limitations to our study include limited generalizability, confounding, and possible reverse causation.

**Data Availability Statement:** The data underlying the results presented in the study are available from the CMS Open Payments database (https://

openpaymentsdata.cms.gov/), the NPPES NPI Registry (https://npiregistry.cms.hhs.gov/), the Physican Compare database (https://www.medicare.gov/physiciancompare/), and Medicare Provider Utilization and the Payments database (https://www.cms.gov/Research-Statistics-Data-and-Systems/Statistics-Trends-and-Reports/Medicare-Provider-Charge-Data/Part-D-Prescriber).

**Funding:** KI was supported by the National Institutes of Health (NIH)/NIDDK grant F99DK126119 and the Burroughs Wellcome Fund Inter-school Training Program in Chronic Diseases (BWF-CHIP). YT was supported by NIH/NIMHD Grant R01MD013913 and NIH/NIA Grant R01AG068633 for other work not related to this study. CMM and OKD were supported by CDC cooperative agreement U18DP006128-04 and NIH/NIDDK grant R18DK105464-04. CMM was also supported by resource Centers for Minority Aging Research Center for Health Improvement of Minority Elderly under National Institutes of Health (NIH)/NIA under Grant P30AG021684, from NIH/National Center for Advancing Translational Sciences UCLA Clinical and Translational Science Institute under Grant UL1TR001881, and the Barbara A. Levey and Gerald S. Levey Endowed Chair in Medicine. CMM is a member of the United States Preventive Services Task Force (USPSTF). This article does not necessarily represent the views and policies of the NIH and USPSTF. Any study sponsors were not involved in study design, data collection and analysis, interpretation, writing, or the decision to submit the article for publication.

**Competing interests:** The authors have declared that no competing interests exist.

**Abbreviations:** CMS, Centers for Medicare and Medicaid Services; NPI, National Provider Identifier; NPPES, National Plan & Provider Enumeration System; OLS, ordinary least squares.

## Conclusions

Industry marketing payments to physicians for long-acting insulin were associated with the physicians' prescriptions and costs of long-acting insulin in the subsequent year. Future research is needed to assess whether policy interventions on physician–industry financial relationships will help to ensure appropriate prescriptions and limit overall costs of this essential drug for diabetes care.

## Author summary

### Why was this study done?

- The financial relationships between physicians and pharmaceutical companies have received increased attention due to their potential to influence clinical decision-making.
- Although Medicare spending on insulin increased 840% over the last decade, empirical evidence about whether the physician–industry financial relationship contributes to physicians' prescriptions and costs of long-acting insulin, the most commonly used and most costly type of insulin, is lacking.
- To address this knowledge gap, we aimed to investigate the association between physicians' receipt of industry payments and their subsequent prescription of long-acting insulin in the United States.

### What did the researchers do and find?

- Among 145,587 physicians treating Medicare beneficiaries included in our study by linking databases of the Open Payments and the Medicare Part D, 51,851 physicians received industry payments for long-acting insulin worth $22.3 million.
- Using the propensity score–matched analysis adjusting for physician characteristics, we found that physicians who received the payments prescribed a higher number of claims, higher costs for total claims, and higher costs per claim of long-acting insulin, compared with physicians who did not receive the payments.
- The association was also found for changes in these outcomes from 2016 to 2017.
- Meals constituted 96% of the total number of industry payments for long-acting insulin, and we found the dose–response relationship between the number of meals received in 2016 and the utilization of long-acting insulin in 2017.

### What do these findings mean?

- Our findings indicate that industry marketing related to long-acting insulin, even those at relatively low dollar amounts such as meals, may influence physicians' prescriptions of long-acting insulin in clinical practice.

- Future research is needed to investigate whether policy interventions on the financial relationships between physicians and industry are effective to ensure appropriate prescriptions and save overall costs of long-acting insulin.

## Introduction

Financial relationships between physicians and the pharmaceutical companies have received increased scrutiny, because while industry marketing often provides physicians with important updates on new research evidence, clinical guidelines, and new treatments, such relationships may also influence physicians' prescription practices [1,2]. As a part of the Affordable Care Act, since 2013, the Open Payments program has mandated that medical product manufacturers publicly report all payments to physicians and teaching hospitals [3]. Previous studies have documented associations between industry payments and prescriptions for specific drugs including opioids [4,5] and cardiovascular drugs [6,7]. Some studies also showed that even the receipt of meals—the most frequent and inexpensive type of industry payments [8]—was associated with an increased rate of prescriptions of drugs from the sponsoring manufacturer [7]. Improving awareness about such associations for long-acting insulin analogs is needed because they are more expensive than other oral antihyperglycemic therapies and human insulin, which could be alternative options for some patients with type 2 diabetes [9,10], and unnecessary prescriptions of long-acting insulins that are not supported by clinical guidelines [11] should be avoided to reduce the risk of hypoglycemia and save costs. However, little is known as to whether the financial relationship between physicians and industry is associated with an increased rate of physicians' prescriptions of long-acting insulin.

A study specific to insulin is important because Medicare Part D spending on insulin increased 840% from $1.4 billion in 2007 to $13.3 billion in 2017 [12,13], mainly due to the rapidly increasing trajectory of insulin prices and the number of Medicare beneficiaries using insulin in the United States [13–16]. In response to the increased spending and limited affordability of this expensive but essential drug for diabetes care, the Centers for Medicare and Medicaid Services (CMS) recently announced a voluntary model under which some Part D plans and pharmaceutical manufacturers would come together to lower Medicare beneficiaries' out-of-pocket cost for insulin to $35 per month, starting January 1, 2021 [17]. In addition, pharmaceutical companies have recently started marketing follow-on biologics that are less expensive and similar types to their brand-name insulin products [18]. Since physicians are the conduit between insulin suppliers and patients with diabetes, it is vital to understand the dynamics of insulin prescribing within a national sample of physicians. Specifically, it is critically important to investigate whether the financial relationship between physicians and pharmaceutical companies is one of the upstream determinants of physicians' prescription of long-acting insulin—the most commonly marketed, commonly used, and most costly type of insulin [9,19,20].

Therefore, using physician databases from the Open Payments program and Medicare Part D, we examined the association between physicians' receipt of industry payments and their subsequent prescription of long-acting insulin. To assess whether any association varies based on the number of industry marketing encounters, particularly in the form of meals, we also examined the dose–response relationship between the number of meals sponsored by the pharmaceutical companies and prescriptions of long-acting insulin.

## Methods

### Data sources

This study linked 4 publicly available databases: (i) the CMS Open Payments data for 2016 [21]; (ii) the CMS National Plan & Provider Enumeration System (NPPES) database [22]; (iii) the CMS Physician Compare database [23]; and (iv) the CMS Medicare Provider Utilization and Payment database for 2017 [24]. To address temporality, we used industry payments data in 2016 and prescription data in 2017. The prespecified analysis plan is described in **S1 Text**.

First, using exact matching of physicians' full name and zip code, we linked the Open Payments database that contains physician-level data on industry payments they received with the NPPES database that contains physicians' National Provider Identifier (NPI) as previous studies did [8,25]. Under the Physician Payment Sunshine Act, biomedical companies are required to report the information about payments to physicians to CMS, and physicians are encouraged to review the information before publication and dispute submitted reports if correction is needed [21]. Using physicians' NPI, the merged database of the Open Payments and NPPES databases was linked to the Physician Compare database that contains physician characteristics (i.e., sex, years in practice, specialty, and medical school attended) and the Medicare Provider Utilization and Payment database that contains physician-level data on drug utilization (e.g., the number of claims including refills, the aggregate cost paid for all claims, etc.). To ensure that physicians included in this study have the potential to prescribe long-acting insulin, we restricted to those who prescribed at least 1 antihyperglycemic therapy in both 2016 and 2017. Further details regarding the derivation of the study population are provided in **S1 Fig**.

### Payment data

We identified and extracted all non-research payments for long-acting insulin in the Open Payments database in 2016. The categories of payments include food and beverage (i.e., meals), travel and lodging, speaker compensation or honoraria, and others (e.g., consulting fees, gifts, educational materials, etc.). When a single payment was associated with multiple drugs (up to 5), we divided the amount and number of the payment equally by the number of reported drugs.

### Prescription data

Our primary outcomes were (i) the total claims of long-acting insulin in 2017 (30-day standardized, including refills); (ii) the total costs paid for claims of long-acting insulin in 2017; and (iii) the costs per claim of long-acting insulin in 2017, calculated by dividing total costs by the number of claims. Our secondary outcomes were changes in these 3 outcomes from 2016 to 2017. To protect the privacy of Medicare beneficiaries, records with fewer than 11 claims were not included in Medicare Provider Utilization and Payment Data [24].

### Physician characteristics

We extracted data on physicians' sex, years in practice (estimated from years since graduation of medical school), and the medical school graduated from the Physician Compare database. We categorized medical schools into 3 groups according to the 2017 U.S. News & World Report, which lists the research ranking of US medical schools in 2017 [26]: ranked 1 to 20, 21 to 50, and others (including all unranked and foreign medical schools). Physicians' specialty was classified into primary care physicians, endocrinologists, cardiologists, nephrologists, pulmonologists, infectious disease physicians, allergy/immunology physicians or rheumatologists,

gastroenterologists, surgeons, and other specialties, using taxonomy codes from the NPPES databases.

## Propensity score matching

We employed nearest-neighbor propensity score matching without replacement to 1:1 match physicians who received industry payments for long-acting insulin with those who did not receive these payments. We used a logistic regression model to derive the propensity scores for the receipt of payments in 2016 that included physician characteristics (sex, years in practice, medical school attended, and specialty) which were considered to be associated with the receipt of industry payments and prescriptions. We used a caliper of 0.20 standard deviation of the logit of the propensity score and evaluated the balance between the 2 comparison groups using the standardized mean difference across covariates (<10% indicated successful balance) [27].

## Statistical analysis

We described baseline physician characteristics before and after propensity score matching according to whether they received industry payments for long-acting insulin in 2017. Then, after confirming that the matching balanced potential confounders, we used a paired *t* test to compare matched pairs of physicians with regard to the total claims, the total costs, and the costs per claim for long-acting insulin in 2017. We also conducted a paired *t* test for the change in these outcomes from 2016 to 2017. Finally, because meals are the most common and inexpensive type of industry payments, we examined the association between the number of meals received in 2016 (as continuous; per 5 meals to calculate interpretable estimates) and the total claims, the total costs, and the costs per claim for long-acting insulin in 2017, using the entire cohort prior to propensity score matching. Given the possible nonlinear relationship, we reanalyzed the data using the 6 categories of the number of meals received in 2016 (0, 1, 2 to 5, 6 to 10, 11 to 15, and 16+) instead of continuous variables. In these analyses, we employed multivariable negative binomial regression models adjusting for the same covariates used in the propensity score matching. The trend test was conducted using the median number of meals received in each category as a continuous variable.

## Sensitivity analyses

Given the possible bias due to residual confounding even after propensity score matching, we reanalyzed the propensity score–matched samples using ordinary least squares (OLS) regression models with Huber–White robust standard errors adjusting for the covariates used to estimate the propensity scores. In addition, to address the possible model misspecification (i.e., to account for the potential right-skewed distributions of residuals), we also conducted the analysis using negative binomial regression models adjusting for the same variables.

## Additional analyses

Using the entire cohort prior to propensity score matching, we conducted the following 3 additional analyses. First, to address the potential reverse causality issue (e.g., pharmaceutical companies might have targeted physicians with a large number of prescriptions of antihyperglycemic therapies), we additionally adjusted for the total number of Medicare Part D claims of antihyperglycemic therapies in 2016. Second, as physicians who prescribed long-acting insulin before 2017 might have been more likely to receive industry payments for long-acting insulin in 2016, we reanalyzed the data restricting physicians to those who did not prescribe

long-acting insulin in 2016. Third, given the potential different distribution of claims and costs across long-acting insulin products (i.e., Lantus, Levemir, Toujeo, Tresiba, and Basaglar), we investigated the association between the receipt of industry payments and prescriptions for each insulin product.

All analyses were performed using Stata, version 15 (Stata Corp, College Station, Texas, United States of America). After fitting regression models, we calculated the predictive value of each outcome under hypothetical exposure levels with the observed distribution of covariates using *margins command* in Stata [28]. The study was exempted by the institutional review board at the University of California, Los Angeles (IRB#18–001960).

## Results

### Physician characteristics

Physician characteristics are shown in **Table 1**. Physicians who received payments in 2016 were more likely to be male, endocrinologists, and have more years in practice. They were less likely to have graduated from 1 to 50 ranked US medical schools. The 2 treatment groups of

**Table 1. Physician characteristics who prescribed antihyperglycemic therapies in 2016 and 2017 according to whether they received industry payments for long-acting insulin in 2016.**

| | Cohort before propensity matching[a] | | | Cohort after propensity matching[a] | | |
|---|---|---|---|---|---|---|
| | Physicians who received payments | Physicians who did not receive payments | Standardized difference, % | Physicians who received payments | Physicians who did not receive payments | Standardized difference, % |
| **Number of physicians** | 51,851 | 93,736 | | 51,295 | 51,295 | |
| **Sex, *N* (%)** | | | | | | |
| Female | 32.0 | 39.1 | −14.9 | 32.4 | 32.1 | 0.7 |
| Male | 68.0 | 60.9 | 14.9 | 67.6 | 67.9 | −0.7 |
| **Years in practice, mean** | 25.2 | 23.8 | 11.2 | 25.0 | 25.1 | −0.6 |
| **Specialty, %** | | | | | | |
| Primary care | 92.2 | 87.0 | 16.9 | 93.2 | 93.2 | −0.0 |
| Endocrinology | 4.6 | 2.0 | 14.5 | 3.5 | 3.5 | 0.0 |
| Cardiology | 0.8 | 2.8 | −14.9 | 0.8 | 0.8 | 0.1 |
| Nephrology | 0.7 | 2.1 | −12.2 | 0.7 | 0.7 | −0.1 |
| Pulmonology | 0.3 | 0.7 | −5.2 | 0.3 | 0.3 | 0.2 |
| Infectious disease | 0.2 | 0.8 | −8.2 | 0.2 | 0.2 | 0.1 |
| Allergy/immunology and rheumatology | 0.2 | 0.5 | −6.3 | 0.2 | 0.2 | 0.0 |
| Gastroenterology | 0.2 | 0.5 | −5.5 | 0.2 | 0.2 | 0.0 |
| Surgery | 0.3 | 1.0 | −9.0 | 0.3 | 0.3 | −0.1 |
| Other specialties[b] | 0.6 | 2.6 | −15.8 | 0.6 | 0.6 | 0.0 |
| **Medical school attended[c], %** | | | | | | |
| Top 20 schools in the US | 4.2 | 8.2 | −16.7 | 4.3 | 4.3 | −0.1 |
| US schools ranked 21 to 50 | 13.1 | 16.7 | −10.0 | 13.3 | 13.3 | 0.1 |
| Other schools | 82.7 | 75.1 | 18.6 | 82.5 | 82.5 | 0.0 |

[a]One-to-one pair matching was performed by nearest-neighbor matching without replacement, with the use of a caliper width equal to 0.2 of the standard deviation of the logit of the propensity score. A standardized difference of less than 10.0% was considered to indicate a negligible imbalance between the 2 groups.

[b]Other specialties include hematology/oncology, neurology, ophthalmology, dermatology, radiology, anesthesiology, etc.

[c]Top 20 schools and Top 21 to 50 schools in US were defined based on the U.S. News & World Report ranking (research ranking) in 2017. Other schools include foreign medical schools.

physicians were well balanced on all baseline covariates after propensity score matching (i.e., standardized differences were <1% for all covariates). The distribution of the propensity scores across the 2 treatment groups is shown in **S2 Fig**.

Among 145,587 physicians who prescribed at least 1 antihyperglycemic therapy for Medicare beneficiaries in 2016 and 2017 and had physician-level information (i.e., our entire study sample before propensity score matching), industry payments for long-acting insulin worth $22.3 million were made to 51,851 physicians in 2016. Food and beverages were the most common type of payments (96%), and speaking fees constituted the largest value ($12,827,441) (**Table 2**). In 2017, 13.3 million claims with a total cost of $4.7 billion were filed for long-acting insulin by 102,960 physicians (71% of 145,587) caring for Medicare beneficiaries included in this study (**S1 Table**).

### Industry payments in 2016 and prescription in 2017

Among 102,590 propensity score–matched physicians, we found that physicians who received industry payments for long-acting insulin in 2016 were associated with a larger number of total claims in 2017 (adjusted difference, 57.8; 95% CI, 55.8 to 59.7), higher costs paid for total claims in 2017 (adjusted difference, +$22,111; 95% CI, +$21,387 to +$22,836), and higher costs per claim (adjusted difference, +$71.1; 95% CI, +$69.0 to +$73.2) of long-acting insulin in 2017, compared with physicians who did not receive payments in 2016 (**Table 3**). We also found the association between the receipt of payments and change in the following 3 outcomes from 2016 to 2017: change in claims (adjusted difference, 2.4; 95% CI, 1.8 to 3.0), change in costs paid for total claims (adjusted difference, +$1,209; 95% CI, +$976 to +$1,442), and change in costs per claim (adjusted difference, +$2.4; 95% CI, 0.6 to 4.1) (**Table 4**). We observed a reduction of costs per claim of long-acting insulin from 2016 to 2017 among both groups of physicians.

We found the association between the number of industry-sponsored meals for long-acting insulin received in 2016 (per 5 meals) and the total claims (adjusted relative ratio [95% CI] = 1.241 [1.235 to 1.246]), the total costs (adjusted relative ratio [95% CI] = 1.268 [1.262 to 1.273]), and the costs per claim (adjusted relative ratio [95% CI] = 1.094 [1.090 to 1.099]) for long-acting insulin in 2017, after adjusting for physician characteristics. We found the consistent results for all outcomes (p for trend <0.001) when using 6 categories of the number of industry-sponsored meals for long-acting insulin received in 2016 (**Fig 1**).

### Sensitivity analyses

The results did not substantially change when we reanalyzed the propensity score–matched samples using the OLS regression model (**S2 Table**) and the negative binomial regression model (**S3 Table**), adjusting for covariates used to estimate the propensity score.

**Table 2. Characteristics of payments involving long-acting insulin in 2016.**

| Nature of payments | Number of payments[a] | Number of physicians | Total payment amount, $ | Median value of the payment, $ (IQR) |
|---|---|---|---|---|
| Food and beverages | 427,841 | 51,851 | $7,030,045 | $15 (12 to 18) |
| Travel and lodging | 6,398 | 579 | $1,653,082 | $197 (102 to 320) |
| Speaking fees or honoraria | 6,458 | 645 | $12,827,441 | $1,880 (1,449 to 2,227) |
| Others[b] | 4,093 | 2,726 | $801,289 | $5 (4 to 14) |

[a]The proportion of payments for single drug and multiple drugs were 67% and 33%, respectively. Only 0.1% of payments were related to 5 products.

[b]Others include education, grant, gifts, entertainment, and space rental or facility fees.

IQR, interquartile range.

**Table 3. Association between the receipt of industry payments for long-acting insulin in 2016 and claims of long-acting insulin in 2017 adjusting for physician characteristics.**

| | Physicians who received industry payments for long-acting insulin in 2016 | Physicians who did not receive industry payments for long-acting insulin in 2016 | P value |
|---|---|---|---|
| **Claims of long-acting insulin in 2017** | | | |
| Mean (95% CI) | 134.5 (133.0 to 136.1) | 76.8 (75.6 to 77.9) | <0.001 |
| Difference (95% CI) | 57.8 (55.8 to 59.7) | | |
| **Costs paid for all claims of long-acting insulin in 2017** | | | |
| Mean (95% CI) | $48,201 (47,606 to 48,797) | $26,090 (25,677 to 26,502) | <0.001 |
| Difference (95% CI) | $22,111 (21,387 to 22,836) | | |
| **Costs per claim of long-acting insulin in 2017[a]** | | | |
| Mean (95% CI) | $300.4 (299.0 to 301.8) | $229.3 (227.8 to 230.9) | <0.001 |
| Difference (95% CI) | $71.1 (69.0 to 73.2) | | |

[a]Estimated by [costs paid for all claims of long-acting insulin in 2017]/[number of all claims of long-acting insulin in 2017]. No claims were replaced as zero.

CI, confidence interval.

## Additional analyses

The association was also observed when we additionally adjusted for the total claims of antihyperglycemic therapies (adjusted difference in claims, 3.9; 95% CI, 2.7 to 5.1; adjusted difference in costs, +$2,820; 95% CI, +$2,348 to +$3,294; adjusted difference in costs per claim, +$52.3; 95% CI, +$50.4 to +$54.3; **S4 Table**) and restricted the analysis to physicians who did not prescribe long-acting insulin in 2016 (adjusted difference in claims, 3.6; 95% CI, 3.1 to 4.2; adjusted difference in costs, +$1,347; 95% CI, +$1,147 to +$1,546; adjusted difference in costs per claim, +$33.1; 95% CI, +$28.8 to +$37.4; **S5 Table**). Across the brands of long-acting insulin, the association was consistently found with the largest absolute difference for Lantus followed by Levemir, and the largest relative ratio for Toujeo followed by Tresiba (**S3 Fig**).

**Table 4. Association between the receipt of industry payments for long-acting insulin in 2016 and change in claims of long-acting insulin from 2016 to 2017 adjusting for physician characteristics.**

| | Physicians who received industry payments for long-acting insulin in 2016 | Physicians who did not receive industry payments for long-acting insulin in 2016 | P value |
|---|---|---|---|
| **Change in claims of long-acting insulin from 2016 to 2017** | | | |
| Mean (95% CI) | 4.4 (3.9 to 4.9) | 2.0 (1.6 to 2.4) | <0.001 |
| Difference (95% CI) | 2.4 (1.8 to 3.0) | | |
| **Change in costs paid for all claims of long-acting insulin from 2016 to 2017** | | | |
| Mean (95% CI) | $1,707 (1,520 to 1,894) | $498 (358 to 637) | <0.001 |
| Difference (95% CI) | $1,209 (976 to 1,442) | | |
| **Change in costs per claim of long-acting insulin from 2016 to 2017[a]** | | | |
| Mean (95% CI) | −$1.2 (−2.4 to −0.6) | −$3.6 (−4.9 to −2.3) | 0.008 |
| Difference (95% CI) | $2.4 (0.6 to 4.1) | | |

[a]Estimated by [costs paid for all claims of long-acting insulin in 2017]/[number of all claims of long-acting insulin in 2017] − [costs paid for all claims of long-acting insulin in 2016]/[number of all claims of long-acting insulin in 2016]. No claims were replaced as zero.

CI, confidence interval.

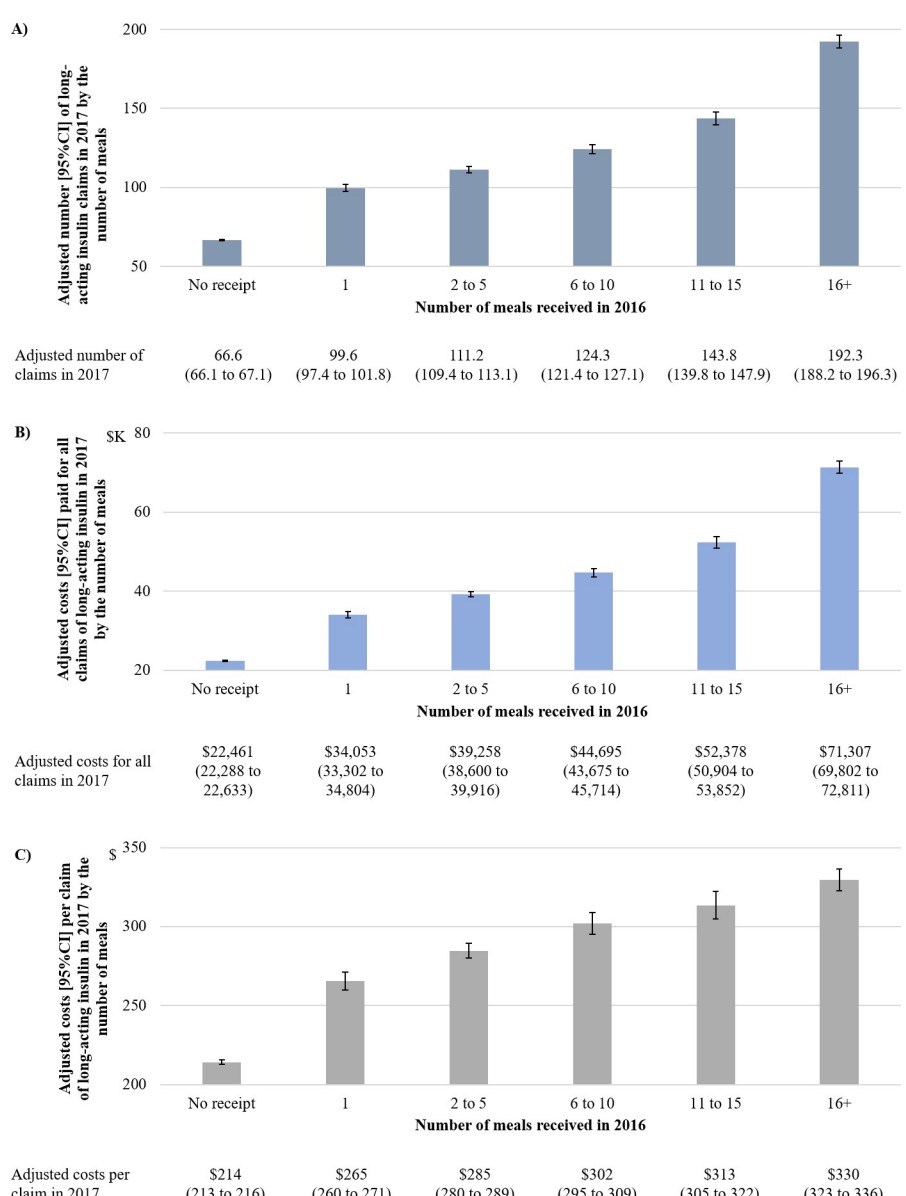

**Fig 1. Association between the receipt of industry payments and claims of long-acting insulin by the number of meals.** Negative binomial regression model was employed to adjust for physicians' sex, years in practice, specialty, and medical school attended for **(A)** number of claims, **(B)** cost paid for all claims, and **(C)** costs per claim of long-acting insulin in 2017 by the number of meals received in 2016.

## Discussion

Using the US national database of physicians treating Medicare beneficiaries, we found that the receipt of industry payments for long-acting insulin in 2016 was associated with increased numbers of claims, costs paid for all claims, and costs per claim of long-acting insulin in 2017. We also found a consistent association for the change in these outcomes from 2016 to 2017. Meals constituted 96% of the total number of industry payments for long-acting insulin, and the number of meals received in 2016 had a dose–response relationship with the utilization of long-acting insulin in 2017.

Prior studies have demonstrated similar findings for antihyperglycemic medications other than insulin. A previous ecological cross-sectional study showed an association between industry payments and regional prescription of marketed non-insulin antihyperglycemic therapies [29]. Another cross-sectional study showed that the receipt of industry payments for the combination drug of saxagliptin and metformin was associated with the increased odds of prescribing this drug [30]. However, these studies used Open Payments data in 2014. The Open Payments program started in 2013, and its data accuracy has improved since 2014 [3], suggesting the need to establish these relationships in updated data. Furthermore, whether these findings also apply to long-acting insulin has not previously been established. In this context, our findings using more recent databases with well-defined temporality (i.e., payments in 2016 and prescription in 2017) and physician-level characteristics, along with propensity score matching, provide new robust evidence about the potential impact of industry marketing on prescriptions of long-acting insulin—a key driver of increased spending in the healthcare system in the US [20].

The influence of the physician–industry financial relationship on clinical practice for diabetes care involving insulin has been of great concern under the regulating system for pharmaceutical manufactures setting insulin prices and direct competition in the insulin market [16,31]. In this context, our findings focusing on long-acting insulin (which constitutes a large proportion of insulin prescriptions and expenditures in the US [20]) generate a hypothesis that physician–industry financial relationship may increase the overall costs of long-acting insulin for patients with diabetes, which works against the national effort to reduce the patients' financial burden of using insulin. Although our data do not adjudicate the appropriateness of the prescription, it may also be possible that industry marketing payments introduce physicians' preference of prescribing long-acting insulin rather than cheaper, oral antihyperglycemic therapies that may be equally effective in some clinical situations, given the prior literature on this topic [4–7]. More investigations are warranted for long-acting insulins to avoid unnecessary prescriptions of this costly and common drugs that affect millions of Americans with diabetes.

Of note, costs per claim paid for long-acting insulin generally decreased from 2016 to 2017 with the larger reduction among physicians who did not receive industry payments. This indicates that there is still an opportunity for physicians whose behavior may be influenced by industry payments to reduce costs for long-acting insulin or at least help hold the line against additional cost increases. In December 2015, a follow-on insulin glargine (Basaglar [Lilly]) received US Food and Drug Administration approval [32]. Although our data in 2017 included only a few physicians who prescribed Basaglar and was not sufficient to make an argument specific to this drug, future studies are needed to understand whether programs without conflicts of interest such as academic detailing [33] or other educational programs focusing on the follow-on insulin are effective to reduce the overall insulin costs by providing unbiased third-party information on long-acting insulins to physicians.

It is noteworthy that we found the significant association between industry payments and utilization of long-acting insulin even when additionally adjusting for total claims of antihyperglycemic therapies or restricting the physician sample to those who did not prescribe long-acting insulin in 2016. Although the magnitude of the association was smaller than our main findings, these results, along with the results from the analyses for change in outcomes from 2016 to 2017, support the presence of direct impact of industry payments on prescriptions and costs of long-acting insulin.

Our findings also suggest that the frequency of industry marketing encounters, even with inexpensive payments such as meals, may contribute to the increased prescriptions and spending for long-acting insulin. A previous study showed that physicians who specifically received

meals were more likely to prescribe the marketed brand-name medications of the pharmaceutical manufacturer that provided the meals [7]. Given that 96% of industry payments for long-acting insulin were meals and almost all physicians who received other types of payments also received meals in the present study, these findings indicate the importance of interventions that refocus even industry marketing encounters with small payments to promote appropriate prescribing that reduces unnecessary spending on long-acting insulin.

### Limitations of the study

Several limitations of our analysis should be considered. As we used the Medicare Part D database, our findings may not be generalizable to non-Medicare beneficiaries. Given the observational study design, it is possible that our models had residual confounding even though we adjusted for several important physician-level variables. In addition, as the databases did not contain patient-level information, it was possible that patients' characteristics such as severity of diabetes and complications might be different between groups of physicians. We did not have access to individual-level data on other forms of marketing that potentially impact physicians' prescribing behavior [34] including funded research, ownership interests, investments in start-up entities, and other physician–industry interactions without direct payment transfers [35]. In the Open Payments database, each payment listed up to 5 products related to the payment, and, thus, we had a possibility of misclassification due to payments related to more than 5 products including long-acting insulin. However, such misclassification, if any, would not change our findings because 99.9% of payments listed less than 5 products in our final analytical sample. Lastly, even though we clarified the temporality by using payments data in 2016 and prescription data in 2017, some physicians might have attended industry events where information about long-acting insulin was provided because they were interested in prescribing specific long-acting insulin medications. In such a scenario, we might have the risk of reverse causation.

### Conclusions

Among physicians treating Medicare beneficiaries in the US, industry marketing related to long-acting insulin in the prior year was associated with higher utilization of long-acting insulin in the subsequent year. The number of meals had a dose–response relationship with prescriptions for long-acting insulin, suggesting that industry marketing encounters, even those at relatively low dollar amounts, may influence physicians' use of long-acting insulin in clinical practice. Future research is needed to investigate whether policy interventions on the financial relationships between physicians and industry will be an effective strategy to ensure appropriate prescriptions of long-acting insulin.

### Ethical approval

The study was exempted from human subjects review by the institutional review board at University of California, Los Angeles.

### Transparency statement

The corresponding author affirms that the manuscript is an honest, accurate, and transparent account of the study being reported; that no important aspects of the study have been omitted; and that any discrepancies from the study as planned (and, if relevant, registered) have been explained.

## Supporting information

**S1 STROBE Checklist. STROBE checklist.**
(DOC)

**S1 Text. Data analysis plan.**
(DOCX)

**S1 Fig. Flow diagram showing derivation of the study population.**
(TIF)

**S2 Fig. The distribution of the propensity scores across the 2 treatment groups.**
(TIF)

**S3 Fig.** Association between the receipt of industry payments in 2016 and **(A)** the number of claims or **(B)** the cost in 2017 for each long-acting insulin.
(TIF)

**S1 Table. Number (%) of physicians who prescribed long-acting insulin in 2016 and 2017.**
(DOCX)

**S2 Table. Association between the receipt of industry payments for long-acting insulin in 2016 and claims of long-acting insulin in 2017 using OLS regression model adjusting for physician characteristics. OLS, ordinary least squares.**
(DOCX)

**S3 Table. Association between the receipt of industry payments for long-acting insulin in 2016 and claims of long-acting insulin in 2017 using negative binomial regression model adjusting for physician characteristics.**
(DOCX)

**S4 Table. Association between the receipt of industry payments for long-acting insulin in 2016 and claims of long-acting insulin in 2017 using OLS regression model additionally adjusting for total claims of antihyperglycemic therapy in 2016. OLS, ordinary least squares.**
(DOCX)

**S5 Table. Association between the receipt of industry payments for long-acting insulin in 2016 and claims of long-acting insulin in 2017 using OLS regression model adjusting for physician characteristics, restricting physicians who did not prescribe long-acting insulin in 2016.** OLS, ordinary least squares.
(DOCX)

## Author Contributions

**Conceptualization:** Kosuke Inoue, Yusuke Tsugawa, Carol M. Mangione, O. Kenrik Duru.

**Data curation:** Kosuke Inoue, Yusuke Tsugawa.

**Formal analysis:** Kosuke Inoue, Yusuke Tsugawa.

**Funding acquisition:** Yusuke Tsugawa, Carol M. Mangione, O. Kenrik Duru.

**Investigation:** Kosuke Inoue, Yusuke Tsugawa, Carol M. Mangione, O. Kenrik Duru.

**Methodology:** Kosuke Inoue, Yusuke Tsugawa, Carol M. Mangione, O. Kenrik Duru.

**Supervision:** O. Kenrik Duru.

**Validation:** Yusuke Tsugawa, O. Kenrik Duru.

**Visualization:** Kosuke Inoue.

**Writing – original draft:** Kosuke Inoue, O. Kenrik Duru.

**Writing – review & editing:** Kosuke Inoue, Yusuke Tsugawa, Carol M. Mangione, O. Kenrik Duru.

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
