## [Editor Report · Decision Letter 0]

2 Sep 2020

Dear Dr Inoue, 

Thank you for submitting your manuscript entitled "Association Between Industry Payments and Prescriptions of Long-acting Insulin: An Observational Study with Propensity-Score Matching" for consideration by PLOS Medicine.

Your manuscript has now been evaluated by the PLOS Medicine editorial staff as well as by an academic editor with relevant expertise, and I am writing to let you know that we would like to send your submission out for external peer review.

Kind regards,

Caitlin Moyer, Ph.D.

Associate Editor

PLOS Medicine

---

## [Decision Letter · Decision Letter 1]

7 Dec 2020

Dear Dr. Inoue,

Thank you very much for submitting your manuscript "Association Between Industry Payments and Prescriptions of Long-acting Insulin: An Observational Study with Propensity-Score Matching" (PMEDICINE-D-20-04080R1) for consideration at PLOS Medicine. 

Your paper was evaluated by a senior editor and discussed among all the editors here. It was also discussed with an academic editor with relevant expertise, and sent to independent reviewers, including a statistical reviewer (r#1). The reviews are appended at the bottom of this email and any accompanying reviewer attachments can be seen via the link below:

[LINK]

In light of these reviews, I am afraid that we will not be able to accept the manuscript for publication in the journal in its current form, but we would like to consider a revised version that addresses the reviewers' and editors' comments. Obviously we cannot make any decision about publication until we have seen the revised manuscript and your response, and we plan to seek re-review by one or more of the reviewers. 

We expect to receive your revised manuscript by Dec 28 2020 11:59PM. Please email us (plosmedicine@plos.org) if you have any questions or concerns.

We look forward to receiving your revised manuscript. 

Sincerely,

Emma Veitch, PhD

PLOS Medicine

On behalf of Caitlin Moyer, PhD, Associate Editor, 

PLOS Medicine

plosmedicine.org

*At this stage, we ask that you include a short, non-technical Author Summary of your research to make findings accessible to a wide audience that includes both scientists and non-scientists. The Author Summary should immediately follow the Abstract in your revised manuscript. This text is subject to editorial change and should be distinct from the scientific abstract. Please see our author guidelines for more information: https://journals.plos.org/plosmedicine/s/revising-your-manuscript#loc-author-summary

*In the last sentence of the Abstract Methods and Findings section, please include a note about any key limitation(s) of the study's methodology.

*Please clarify whether the analytical approach followed here corresponds to one laid out in a prospective protocol or analysis plan? Please state this (either way) early in the Methods section.

*Because papers are repaginated for publication, please update the STROBE checklist using section and paragraph numbers, rather than page numbers. The authors might also want to consider using instead the RECORD guideline, if they feel that is appropriate (https://www.equator-network.org/reporting-guidelines/record/) as this was designed to support reporting of studies done using routinely collected health data (and/or registry linkage methods, which seem to have been used in your study).

*Due to PLOS Medicine policy/style, please remove trademark symbols from the manuscript and supporting information files.

Comments from the reviewers:

Reviewer #1: 

Thanks for the opportunity to review your manuscript. My role is as a statistical reviewer so my comments are focused on the study design, data, and analysis. This is an interesting manuscript that has made good use of some routinely collected healthcare data. I have put overall comments and queries first and then followed these by queries relating to specific parts of the manuscript (with a page and paragraph reference).

I didn't feel as though there was enough background given to clarify the ramifactions of a switch to long-term insulin for both patients and health systems. My understanding (as a non-expert who has worked with similar data) is that long-term insulin could in fact translate to better outcomes for patients (and also potentially worse outcomes). Is a switch to long-term insulation always inappropriate with current guidelines? Is it possible that although costs are higher, this translates into appropriate expenditure for the improvement in patient outcomes? 

My other main query was the interpretation of the results - in the main results the difference in claims (134 vs 77) and costs ($48k vs $26k) between the two groups is quite stark. When adjusted for total claims of antihyperglycemic medicines the difference diminishes (93.4 vs 89.5; and $33k vs $31k). This needs further discussion - this would seem to imply that the main mechanism for the increases in claims and costs comes from the industry funding influencing overall activity around antihyperglyemic therapy which doesn't seem to make sense on face value. Could patient load (i.e. number of patients with diabetes per physician) be on the causal pathway here and not adjusted for during the matching process and leading to the discrepancy between the main results and this particular sensitivity analysis?

P6. Paragraph 2. It wasn't clear that direct matching (i.e. exact matching of the names and ZIP) was used even after checking the two references - could you just add whether it was exact matching or a details of statistical linkage key if the info was used this way? 

Does the MPUP database record medication dispensings or prescriptions? 

P7, Paragraph 1. What proportion of the payments were single-drug, and what proportion associated with a multi-drug record? Were payments for more than five drugs excluded at this step or was 5 the limit that payments were adjusted for?

P8. Did you use the 0.2 SD of the logit of the PS, or PS on the original scale? Where is the balance information? Did you include interactions between these variables? 

What was the process of considering which physician characteristics were considered to be important confounders of the association between industry payments and long-term insulin prescribing?

Was there any missing data in the variables used to create the propensity scores? If so, how was this dealt with?

What was the criteria used for the categories in the trend test? Why was this used over the original value of the meals received variable? 

P9. Sensitivity analyses. I assume that the negative binomial sensitivity analysis was used as cost data often has an inconvenient distribution (and this should be clarified in the methods), was the glm with sandwich errors (H-W) a way to estimate a model with a normal distribution while accounting for the matching? 

P10, Methods. Was psmatch2 or teffects used for the matching? 

Was common support assessed? A figure of PS distribution by exposure group would be a helpful addition to the appendix.

 What method was used to get the estimates of claims, costs etc. per exposure group? Margins in Stata? Was this set at particular levels of the covariates, or at the average for the entire cohort? 

P10, Results, Paragraph 1. Is it worth adding that the standardised differences after matching were below the target (and in fact are all below 1% different?)

P11/P12. Is it possible (given space limitations) to report the key effect estimates from the sensivity analyses in the results? There is still a difference between the two exposures groups when there is adjustment for total claims of antihyperglycemic therapies but the magnitude of the difference is greatly reduced. 

S1. Figure 1. This is an important diagram understanding how the data for the study was created. To clarify, there were ~5,000 physicians in the open payments database who weren't able to be matched to the NPPES data? Similarly, ~43,000 providers who had prescribed at least one antihyperglycaemic in 2016/2017 weren't matched to the physician Compare database? Also, there were ~60228 physicians who had received payments but then only 51,851 that had been classed as having done this in the final database - were these physicians that couldn't be linked with the NPI?

 I appreciate that this is already a detailed diagram but if there is a way to include information on the numbers excluded at each step this would be very helpful, with linked administrative data the quality of the linkage process is very important for study validity and this would help me assess this aspect of the study. 

Reviewer #2: 

This manuscript investigates the association between payments to physicians and prescribing of long-acting insulin. It adds to the growing literature showing that industry payments have an influence on the way that doctors prescribe. My comments are of a relatively minor nature:

1. Besides the increase in the price of insulin Medicare spending could also have increased because of more patients using insulin.

2. Page 4, 2nd paragraph, 6th line: The correct word should be "companies" not "industries".

3. Rather than using references 13-15 for the influence of promotion on physicians' prescribing behaviour, the authors should cite the systematic review by Spurling et al in PLoS Medicine that was published in 2010.

4. Besides academic detailing what other types of policy interventions are the authors proposing to regulate financial interactions between physicians and the pharmaceutical industry?

Joel Lexchin

Reviewer #3: 

This study examines a potentially interesting issue—the association between industry payments and prescription of long-acting insulin. The authors do a good job of situating the study in relation to prior work in this area. As they rightly note, while there have been a number of studies that explore and find associations between industry payments and prescribing in other areas of medicine, relatively little is known about these dynamics in diabetes care. A few similar studies of diabetes care have relied on older (2014) data. Overall, then, the study has the potential to make an incremental but meaningful addition to the literature on financial COI and their implications for medicine. 

However, I have two broad concerns about the framing and conceptualization of the study.

First, the authors introduce and motivate the paper with a concern about the rising cost of insulin in the US. This is well-established and a genuine policy problem. But they seem to suggest—implicitly throughout and explicitly at times—that the rising cost of insulin is influenced by physician-industry relationships and the lack of transparency around these relationships. For instance, in the Discussion, they write that "Key factors related to the rise in insulin prices over the past two decades include…the lack of transparency in such financial relationships between physicians and industry for insulin." On the next page, they appear to fault the ADA Working Group for failing to mention "the industry marketing payments asfactors in the escalating cost of insulin." 

However, the authors do not propose a mechanism by which industry payments to physicians, or the higher rates of prescribing that they might induce, would lead to higher drug prices. As such, the findings presented in the study, which show an association between industry payments and prescribing, do not appear to be plausibly linked to the policy problem of interest, namely, the rising cost of insulin. In this sense, the bookends of the paper seem to be dealing with a different problem (i.e., the cost of insulin) than the heart of the paper (i.e., determinants of insulin prescribing). The author should either reframe the Introduction and Discussion sections, or else specify the mechanism by which they believe industry payments to physicians affect insulin prices. 

Second, and somewhat relatedly, it isn't clear how we ought to interpret the implications of higher rates of long-acting insulin prescribing. Typically, in these sorts of studies, investigators are interested in the associations between industry payments and prescribing because of an underlying concern that payments might influence physicians to prescribe in ways that are clinically suboptimal or clinically equivalent but costlier. So, for instance, in DeJong et al. (JAMA IM 2016)—which this manuscript cites—the authors find that industry payments are associated with higher rates of prescribing brand name drugs when generics are available. 

Do higher rates of long-acting insulin prescribing raise similar concerns? It isn't clear from the manuscript. One concern could be that physicians are prescribing an expensive drug, long-acting insulin, when a cheaper alternative would do. But in the Discussion, the authors say that "long-acting insulin is an essential drug and cannot be easily replaced for many patients" so that would seem not to be the case. Later, however, they say that detailing programs "focusing on a less expensive version of long-acting insulin" may be appropriate. If such alternatives exist, that should be stated earlier in the paper. More importantly, if there are both expensive and cheap versions of long-acting insulin, then this study's design would seem to be seriously flawed. In that case, the research question should be whether receipt of industry payments is associated with prescription of the more expensive type of long-acting insulin. But the question that this actually asked here is whether receipt of industry payments is associated with higher rates of prescribing any long-acting insulin.

I suppose another reason to be concerned about higher rates of long-acting insulin prescribing could be that physicians are overprescribing insulin. This is the underlying concern, for example, in recent studies of the association between industry payments and opioid prescribing. But that seems implausible in the case of insulin, and the authors provide no evidence to that effect.

Perhaps some of these issues involving the availability of clinical alternatives and the implications of higher rates of long-acting insulin prescribing are obvious to specialists, but as this is a general medical journal, they would need to be more clearly spelled out.

Overall, these two issues—1) that the authors don't provide a plausible mechanism by which industry payments to physicians drive higher insulin prices, and 2) that the author don't otherwise clarify why higher rates of long-acting insulin are problematic—raise questions about the policy or clinical relevance of the study's findings.

[LINK]

---

## [Decision Letter · Decision Letter 2]

4 May 2021

Dear Dr Inoue, 

On behalf of my colleagues and the Academic Editor, Aaron S Kesselheim, I am pleased to inform you that we have agreed to publish your manuscript "Association Between Industry Payments and Prescriptions of Long-acting Insulin: An Observational Study with Propensity-Score Matching" (PMEDICINE-D-20-04080R2) in PLOS Medicine.

PRESS

Sincerely, 

Dr Raffaella Bosurgi 

Executive Editor 

PLOS Medicine